# Multi-Modal Mobility Morphobot (M4) with appendage repurposing for locomotion plasticity enhancement

Eric Sihite[1], Arash Kalantari[2], Reza Nemovi[1], Alireza Ramezani ⬡[3] ✉ & Morteza Gharib[1]

Robot designs can take many inspirations from nature, where there are many examples of highly resilient and fault-tolerant locomotion strategies to navigate complex terrains by recruiting multi-functional appendages. For example, birds such as Chukars and Hoatzins can repurpose wings for quadrupedal walking and wing-assisted incline running. These animals showcase impressive dexterity in employing the same appendages in different ways and generating multiple modes of locomotion, resulting in highly plastic locomotion traits which enable them to interact and navigate various environments and expand their habitat range. The robotic biomimicry of animals' appendage repurposing can yield mobile robots with unparalleled capabilities. Taking inspiration from animals, we have designed a robot capable of negotiating unstructured, multi-substrate environments, including land and air, by employing its components in different ways as wheels, thrusters, and legs. This robot is called the Multi-Modal Mobility Morphobot, or M4 in short. M4 can employ its multi-functional components composed of several actuator types to (1) fly, (2) roll, (3) crawl, (4) crouch, (5) balance, (6) tumble, (7) scout, and (8) loco-manipulate. M4 can traverse steep slopes of up to 45 deg. and rough terrains with large obstacles when in balancing mode. M4 possesses onboard computers and sensors and can autonomously employ its modes to negotiate an unstructured environment. We present the design of M4 and several experiments showcasing its multi-modal capabilities.

This work aims to design a robot capable of negotiating unstructured, multi-substrate environments with extensive locomotion plasticity by transforming its multi-purpose appendages to achieve different functions, including wheel, leg, and thruster. We call this robot M4, which stands for Multi-Modal Mobility Morphobot (Fig. 1). This morphobot could be used in a broad number of applications, including search and rescue operations, space exploration, automated package handling in residential spaces, and digital agriculture, to name a few.

Envision search and rescue after natural disasters such as earthquakes, flooding, or windstorm (Fig. 2). In the aftermath of unique incidents such as flooding, one event may accompany another that destroys the landscape differently. A hurricane may produce flooding and wind damage to roads and buildings. Or, a landslide may cause the movement of a large rock mass down a slope, dam a river, and create a flood. In these scenarios, M4 can leverage its versatility to achieve mobility that fits diverse mission requirements in search and rescue. For instance, when ground locomotion is not feasible, M4 delivers

[1]Aerospace Engineering Department, California Institute of Technology, 1200 E California Blvd, Pasadena, CA, USA. [2]Jet Propulsion Laboratory (JPL), 4800 Oak Grove Drive, M/S 82-105 Pasadena, CA, USA. [3]Electrical and Computer Engineering Department, Northeastern University, 360 Huntington Ave, Boston, MA, USA. ✉e-mail: a.ramezani@northeastern.edu

**Fig. 1 | Multi-Modal Mobility Morphobot (M4). a** Shows M4 in wheeled mode. **b** Illustrates cartoon depictions of M4's transformation to other modes.

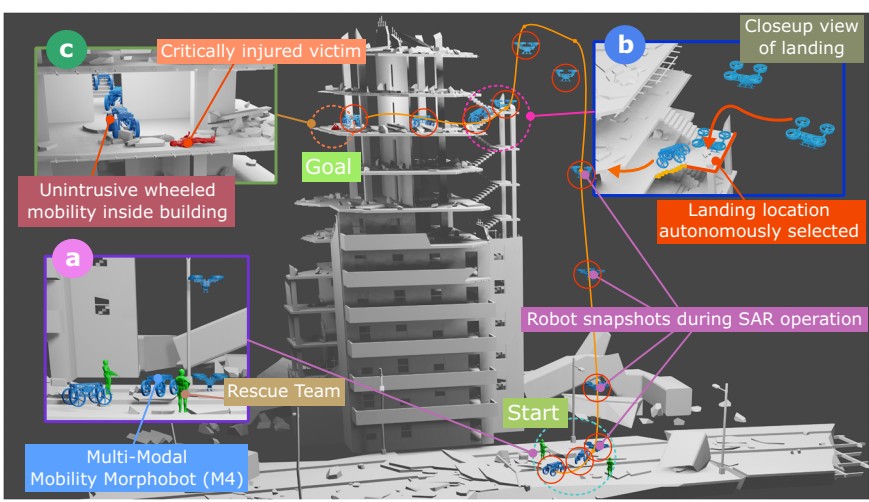

**Fig. 2 | Envisioned search and rescue example. a** An illustration showing the deployment of M4 outside a collapsed multi-story building in the aftermath of an earthquake. **b** M4 employs its aerial mobility to reach quickly and land on inaccessible locations. **c** Other modes, such as wheeled mobility, are employed when a flight is impossible.

critical strategic situational awareness by employing aerial surveying and reconnaissance through multi-purpose scans of the area with a suite of sensors integrated into its design. Aerial mobility inside confined and collapsed buildings is not practical. Imagine mobility inside tight, collapsed stairways and corridors on top floors needed. In that case, M4 utilizes diverse forms of ground locomotion, including four-wheel rolling and crouching, two-wheel rolling and standing (with or without thrusters), quadrupedal walking, or tumbling to negotiate inside collapsed floors. For instance, wheeled and legged mobilities have limitations as they cannot handle rough terrains when obstacles

are larger than the wheels' and legs' size. Instead, M4 tumbles over them, i.e., it leverages the ability to upright using its thrusters to achieve the height advantage needed to fall over large obstacles.

This work presents the design and control of a versatile multi-modal robot called M4 shown in Fig. 1. The contributions of this work are multi-fold. First, we show a significant modal diversity not reported in the literature. Inspired by animals with considerable locomotion plasticity, such as birds, the M4 robot can perform various modes of locomotion by redundancy manipulation through appendage repurposing. M4 repurposes its appendages with its transforming body and

switchable shrouded propellers to switch to an unmanned ground vehicle (UGV), mobile inverted pendulum (MIP), unmanned aerial system (UAS), thruster-assisted MIP, legged locomotion, and loco-manipulation in MIP mode. Second, by repurposing the mobility components in M4, we achieve a scalable design that supports fully autonomous and self-contained operations. We show the robot possesses the payload capacity to carry computers and exteroceptive sensors for fully autonomous multi-modal operations. Third, we combine locomotion diversity and autonomy in M4 to perform novel maneuvers such as tumbling over large obstacles and traveling over steep ramps. This paper presents the mechanical design and the algorithms that enable M4 to perform these modes. These algorithms are explained in the Method Section and entail an optimization-based control (collocation method) and path planning algorithm (multi-modal probabilistic road map [MM-PRM] and A* algorithms). We report the experimental results that substantiate the claimed capabilities.

The overarching objective of the M4 design is to achieve a scalable solution with extensive locomotion plasticity to substantiate the scenarios explained above. We call a mobile robot design scalable if its payload capacity can be increased such that its mobility is not severely affected. While there are various ways to measure scalability, one fundamental approach is to evaluate it based on the maximum allowable payload that the system can carry before it becomes completely immobilized in any mode. Obviously, scalability depends on several factors, including actuators and mechanisms' performance, locomotion modes, and substrate characteristics. Since multi-modal locomotion involves different actuators, mechanisms, modes, and substrates, the scalability problem can be very confounding. For instance, it is generally tough to accommodate the conflicting requirements dictated by ground and aerial locomotion in a single platform. On the one hand, powerful actuators and rugged structures are needed to generate and maintain traction forces or joint torques to successfully realize wheeled or legged locomotion. The plurality of actuators in these systems is very high to substantiate posture control. On the other hand, these actuators and structures are often very bulky, negatively affecting aerial mobility, which depends on light structures.

Here, the question to ask is: Which design views yield scalable robots with large locomotion plasticity? We list three views, including two mainstream views (1–2) that cover the multi-modal designs introduced in literature and one view (3) that has been explored to a very limited extent:

- *View 1: Morpho-Functionality-* In this view, multi-modal locomotion is achieved through body and appendage morphing. These designs comprise manifold rigid (or soft) links and actuated joints that form articulated bodies and appendages. Morphing or shape-shifting is considered the primary mechanism for changing appendage function. The appendages can be, e.g., legs, wings, flippers, wheels, slithering structures, etc., simultaneously by changing their shapes and motions. The transforming body recruits these multi-functional appendages and shares them among different mobility modes.

Many morpho-functional machines with promising morphing designs based on rigid[1–16] and soft bodies[17–23] have been introduced so far. A large number of these designs are legged[7–16], slithering[24], and amphibious[6,25,26] robot. Other unconventional designs such as quadruped with reconfigurable joints[1], transforming robot that can use its wings as legs[27], multi-rotor with morphing body[2], shape-shifting wheeled robot[3,5,9], and adaptive wheel-and-track[4] have been introduced as well. However, these multi-modal robots showcase limited locomotion plasticity (two-three modes)[2–4]. Soft morpho-functional options have been extensively studied too. However, they can accommodate a limited number of modes and have faced scalability challenges. For instance, while soft structures share strong similarities with shape-shifting biological mechanisms in vertebrates and

invertebrates, these engineered elements cannot match their biological counterparts in terms of generated force-motion profile per unit mass[18,20,22]. State-of-the-art soft robots cannot scale up to large, self-contained systems with notable locomotion plasticity since they depend on large accessories such as pneumatic systems or high-voltage power supplies.

- *View 2: Redundancy-* In this view, multi-functionality is achieved by brute-force approaches based on the plurality of appendages that can deliver one function only. Hence, the appendages are not shared among different modes and are fixated on non-morphing bodies. Note that by redundancy we refer to the number of appendages involved in a locomotion mode. We label it redundant if more appendages are required than the minimum number needed for that mode. Therefore, redundancy in actuated joints does not render a system redundant. Consider human bipedal locomotion that consists of two legs each comprising a plural of muscles (analog to robot actuators) that would allow the leg to deliver different functions. In our view, this example is not redundant.

There is a plethora of celebrated works[28–35] that successfully have utilized redundancy in their designs to achieve multi-modal locomotion. These redundant designs present less complexity, which is a benefit, by carrying additional actuators and robotic mechanisms to substantiate legged-aerial[32–34], wheeled-aerial[29,30,35], and amphibious locomotion[28]. For example, the robot designed by ref. 29 is a quad-rotor with wheels and motors affixed at the base of the robot to enable ground mobility for the initially aerial-only robot. Another notable example is *HyTAQ* by ref. 36 which comprises a multi-rotor aerial system encapsulated by a barrel-shaped guard that allows safe wheeled mobility. However, in these designs, there is a strict limit on the number of modes that can be integrated and these robots quickly face added mass issues.

- *View 3: Manipulation of Redundancy by Morphing-* So far, both Views 1–2 with various levels of complexity have been adopted in robotics. In some concepts, morpho-functionality is the main design theme and, in many examples, redundancy. However, in nature, animals showcase a behavior that combines both views; animals utilize their morpho-functional structures to repurpose the appendages to create (or to eliminate) redundancy when needed and gain mobility advantage. For instance, aquatic animals such as turtles and sea lions use their front flippers for swimming. They repurpose the same flippers (Fig. 3a) to support their heavy body weight and to walk on the ground like a quadruped[37]. Or, Meerkats, as shown in Fig. 3b, can eliminate the redundancy in their locomotion apparatuses by standing on their hindlimbs to scout their surroundings. These animals cannot walk well on two legs, but they can use them to elevate their field of view to monitor their surroundings to avoid predators[38].

Birds such as Hoatzins and Chukars manipulate redundancy in their locomotion apparatuses as well. Juvenile Hoatzins showcase wing-assisted walking[39] to move up vertical or steep slopes to refuge and dodge danger (Fig. 3c). They repurpose the wings and shape-shift the articulated body to extract leg functions from their wings and achieve quadrupedal locomotion. Young Hoatzin nestlings retain functional claws in their wings which helps them to manifest quadrupedal locomotion and even climb in the vegetation.

On a similar note, Chukar birds adopt a similar wing repurposing to increase redundancy to support legged locomotion over steep terrain through a phenomenon known as wing-assisted incline running (WAIR)[40] (Fig. 3d). To walk over steep surfaces, they leverage their wings' contributions differently to walk on steep inclinations. Chukar chicks walk and run up steep slopes by beating their developing wings and generating aerodynamic

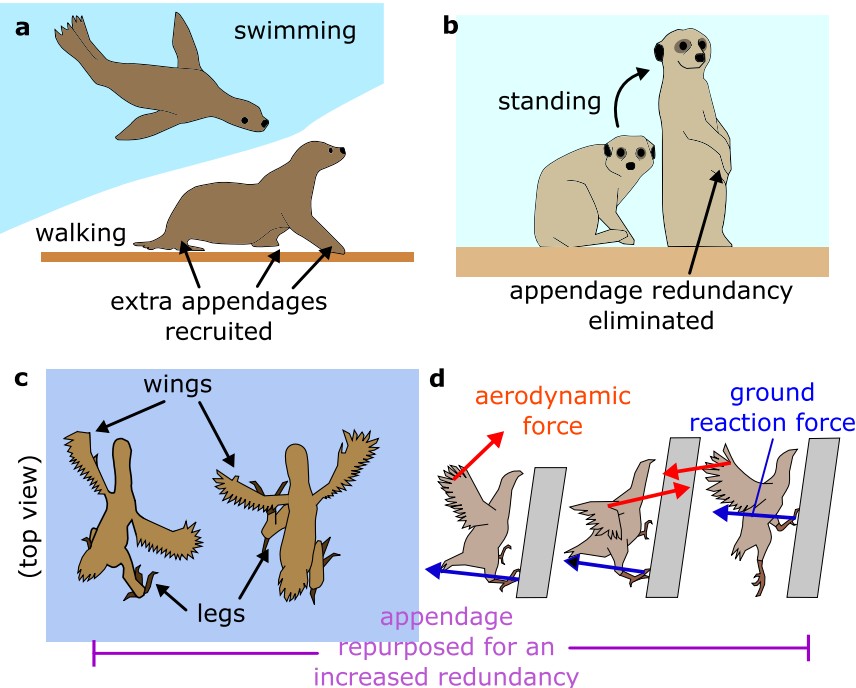

**Fig. 3 | Cartoon depiction of appendage repurposing in different species. a** Sea lions flipper-assisted walking[37]. **b** Meerkats' hindlimb-assisted scouting[38]. **c** Hoatzin nestlings wing-assisted quadrupedal locomotion (Image inspired and modified with permission from[39] authors). **d** Chukar birds' wing-assisted incline walking[40].

lift force which increases the ground contact force at their legs[41]. With the WAIR strategy, mature Chukar birds can negotiate nearly vertical and overhanging slopes as if walking on flat ground.

The robotic biomimicry of these redundancy manipulations through appendage morphing has remained unexplored. The celebrated multi-modal robots presented by refs. 42, 43 possess interesting designs that permit flipper-leg and wheel-leg repurposing to achieve aquatic-legged and wheeled-legged modalities. However, M4 differs from refs. 42, 43, 44 work because M4 exhaust appendage redundancy manipulation through morphing to maximize locomotion plasticity. For instance[42], repurposes four flippers into four legs for walking. Instead, M4 repurposes four legs into:

- Four legs for quadrupedal locomotion (Supplementary Video 1),
- Four thrusters for flight (Supplementary Video 2),
- Two thrusters + two wheels for WAIR over 45-deg slopes (Supplementary Video 3),
- Two thrusters + two wheels for tumble over large obstacles (Supplementary Video 4),
- Two wheels + two hands for loco-manipulation (Supplementary Video 1),
- Two wheels for MIP (Supplementary Video 5),
- Four wheels for UGV (Supplementary Video 1),
- Four wheels for crouching (Supplementary Video 1).

It can be seen that the redundancy manipulation through appendage morphing in M4 is not matched by refs. 42, 43. The extent by which these repurposings are strategized to diversify locomotion modes is very limited in these examples. In addition, in these works, appendage repurposing is not considered as a tool to achieve scalability and combat the conflicting requirements posed by a plurality of locomotion modes. For instance, the MIP maneuver showcased in ref. 43 only works on flat ground and cannot be scaled to steep slopes like M4.

## Results

### Design rationale

By inspecting the state-of-the-art multi-modal robots, we notice that, besides many redundant designs, a large number of soft- and rigid-bodied morphing systems have been introduced so far. By using redundancy and novel adaptive structures, the robotic community has tirelessly worked on democratizing multi-modal robots that can showcase animals' locomotion resiliency and fault tolerance. However, the total number of modes achieved in these examples has remained limited to small numbers. In addition, today's multi-modal robots that face conflicting design requirements are not scalable, i.e., they do not have the payload capacity needed to carry large items to render their multi-modality useful. In these designs, in addition to the added mass from each mode, there is another form of added mass that must be considered to avoid the risk of immobilization. As the mass from other modes adds up, some modes (e.g., UAS and legged modes) require the addition of large actuators, power electronics, and batteries to prevent the risk of immobilization. In other words, in these modes, component size rapidly grows as the total mass increases. Other modes may be less sensitive to mass increase. For instance, the manipulation mode cannot be affected by an increase in the total mass since it depends solely on the object's mass, not the robot's mass. On the contrary, the legged mode is very sensitive to mass increase since joint actuators have to carry the robot's weight.

The main objective of M4 design is to achieve a scalable solution with many mobility modes. Note that, in the design of M4, we are focused on copying animals' strategies to enhance locomotion plasticity rather than mimicking the shape of animal appendages (flapping versus rotary wings). For this objective, we adopt the design approach based on manipulating appendage redundancy through component repurposing for the following reason. This view multi-folds the force-to-weight ratio required for large payloads and demanding locomotion modes through three mechanisms. First, added mass from components is shared by all modes, a key mechanism that motivates appendage repurposing. Second, force amplification becomes possible in a mode through heterogeneous mobility component recruitment. For

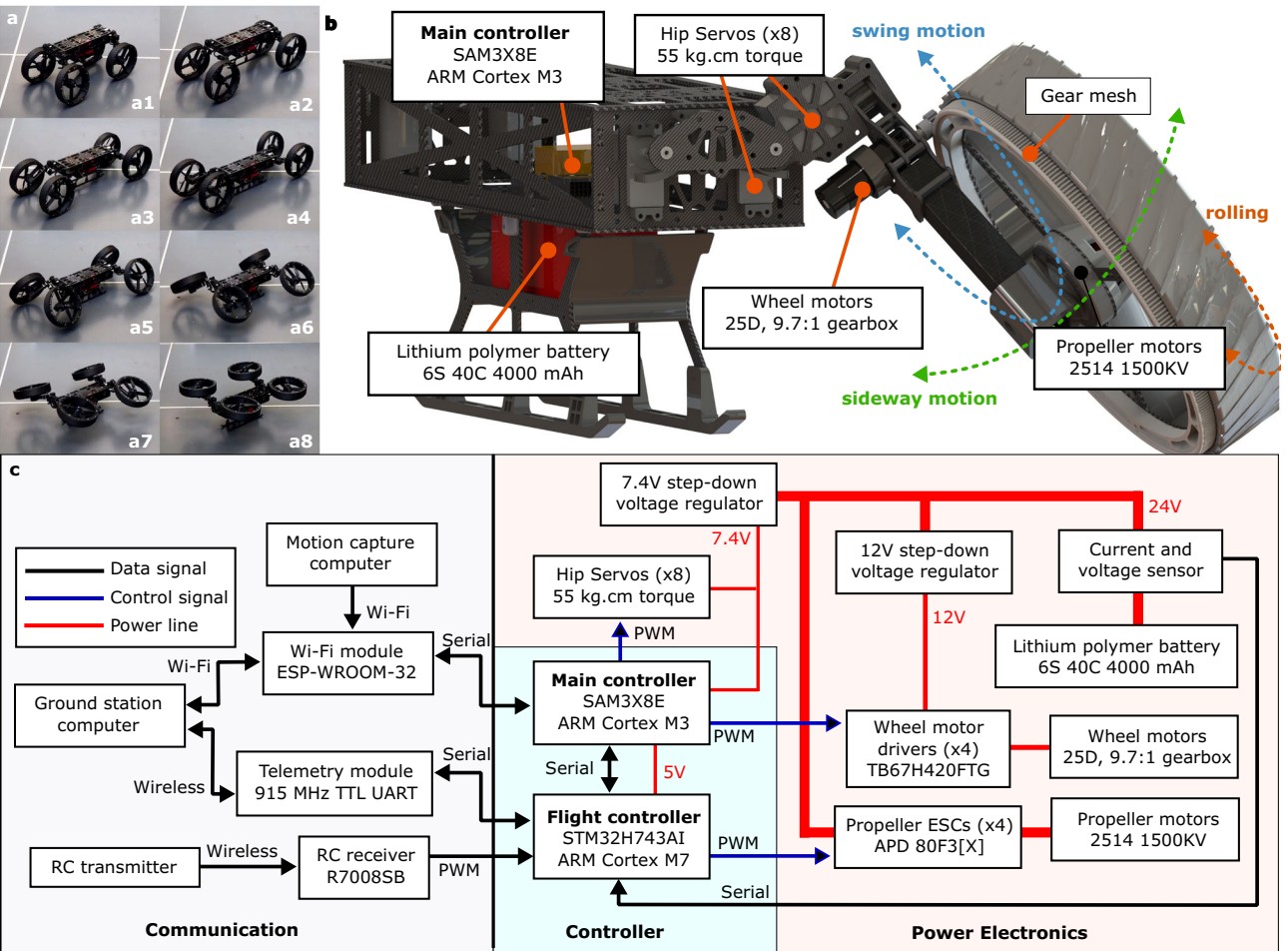

**Fig. 4 | An overview of M4's hardware design. a** Illustrates an example appendage repurposing process in M4 to increase thrust-to-weight ratio fourfold. **b** Shows a closeup view of M4's mechanical design, degrees of freedom, and components. **c** Shows M4's electronics architecture, including the communication, controller, and power electronics components.

instance, aerodynamic lift forces can manipulate contact friction and traction forces in wheeled mobility and allow steep slope locomotion, a behavior inspired by birds' WAIR maneuvers. Third, force amplification becomes possible in a mode through homogeneous mobility component recruitment. For instance, for a fixed mass, the thrust-to-weight ratio doubles and quadruples when switching from UGV to MIP and UAS. To see other benefits of appendage redundancy manipulation that are not explored in M4's design refer to a conceptual design depicted in Supplementary Fig. 1.

## System Overview

The M4 robot, shown in Fig. 4, can switch its modes of mobility between UGV, UAS, MIP, quadrupedal, thruster-assisted MIP, legged locomotion, and manipulation. M4 possesses an articulated body with four legs where each leg has two actuated hip joints for frontal and sagittal leg movements and a shrouded propeller that acts as a wheel and thruster simultaneously. The frontal joints permit the legs to move in the sideway direction. On the other hand, the sagittal joints accommodate forward and backward swing movements in each leg. This body articulation allows various transformations. For instance, as shown in Fig. 4a, to achieve a UAS configuration with a four-fold thrust force, first, the legs swing forward and backward. Then, they turn sideways with the frontal actuators. In M4, the propeller's shroud acts as a wheel which is actuated by a motor that drives through the gears attached to the shroud's rim, as illustrated in Fig. 4b. The propulsion is generated by the propeller and motor inside the shroud aligned with

the wheel axis. If the motion of the propellers and shrouds is considered, the robot possesses a total of 16 actuators and body degrees of freedom (DOF). As a result, the total number of DOFs in M4, including actuated coordinates, body positions, and orientations, is 22.

The mechanical design and components overview of M4 can be seen in Fig. 4b. The robot weighs approximately 6.0 kg with all components, including the onboard computers for low-level control and data collection, sensors (encoders, inertial measurement unit, stereo cameras), communication devices for teleoperation, joint actuators, propulsion motors, power electronics, and battery. M4 measures 0.7 m in length, and 0.35 m in both width and height when in UGV mode. When in MIP mode and dynamically balancing on its two wheels, it is 1.0 m tall, which permits reaching a better vantage point for data collection using its exteroceptive sensors. When in UAS configuration, M4 is 0.3 m tall, and propellers' center points can reach a maximum distance of 0.45 m far apart from each other. Each propeller-motor combination can generate a maximum thrust force of ~2.2 kg-force, therefore reaching roughly 9 kg thrust force in total. Its legs including the wheels are 0.3 m long, and its wheels are 0.25 m in diameter, which allows for traversing bumpy terrain. Table 1 lists the component weight distribution of the most recent M4 design without a stereo camera attached.

The chassis structures and shrouded propeller components in M4 were primarily made of carbon fiber and 3D-printed parts. The 3D-printed parts are fabricated using a fiber-inlay process based on Onyx thermoplastic materials and carbon fiber. These materials were considered due to their great strength-to-weight ratios. M4's system

architecture is outlined in Fig. 4c showing the controller system, power electronics, and communication protocol used in the robot. The robot utilizes two microcontrollers for low-level locomotion control; one is used for posture and wheel motion control, while the other is used to regulate thrusters. In addition to the low-level locomotion controllers, there is a high-level decision-making computer for autonomous multi-modal path planning. The details of M4's dynamic modeling, low-level locomotion controller design, and high-level, multi-modal path planning can be seen in the Methods Section.

## Experimental results

To substantiate the claimed locomotion plasticity in M4, we performed several experiments, including, wheeled locomotion, flight,

MIP, crouching, object manipulation, quadrupedal-legged locomotion, thruster-assisted MIP over steep slopes, and tumbling over large obstacles. In addition, to show M4's design is scalable and can achieve payload capacities that support self-contained operations, we tested fully autonomous multi-modal path-planning using onboard sensors and computers in M4. A summary of these experiments is shown in Figs. 5–8.

Figure 5a shows snapshots of M4 navigating around and over a pond from Supplementary Video 2. M4 is teleoperated (not autonomous) in this test. M4 employs its wheeled mobility to reach the pond's edge, then it transforms into a UAS and flies over the pond to the other side of it. The UGV-UAS transformations follow the steps shown in Fig. 4.

Figure 5b shows the snapshots of the MIP maneuver from Supplementary Video 5. The MIP maneuver was performed in a closed-loop fashion based on the collocation method (see Methods Section). In this experiment, we performed controlled transitions from UGV to MIP and MIP to UGV. In the MIP maneuver, first, the front appendages are repurposed from wheel to thruster by employing the sagittal and frontal joints. Second, the thrusters' force and wheels' tractions are regulated using an optimization-based, nonlinear closed-loop feedback controller in real-time. The body orientation and angular velocity are sensed in real-time, then the control actions are generated to track desired angular rates to achieve a stable MIP configuration. The

**Table 1 | Detailed weight breakdown, total weight = 5.6kg**

| Name | Weight | Name | Weight |
|---|---|---|---|
| Battery (6S 4Ah) | 590 g | Leg assembly (×4) | 400 g |
| Chassis assembly | 940 g | Hip servos (×8) | 560 g |
| Microcontrollers | 115 g | Propeller motors (×4) | 270 g |
| Communication | 120 g | Wheel motors (×4) | 380 g |
| Power electronics | 80 g | Tire assembly (×4) | 1600 g |
| Cables, fasteners, etc. | 440 g | Motor drivers (×4) | 107 g |

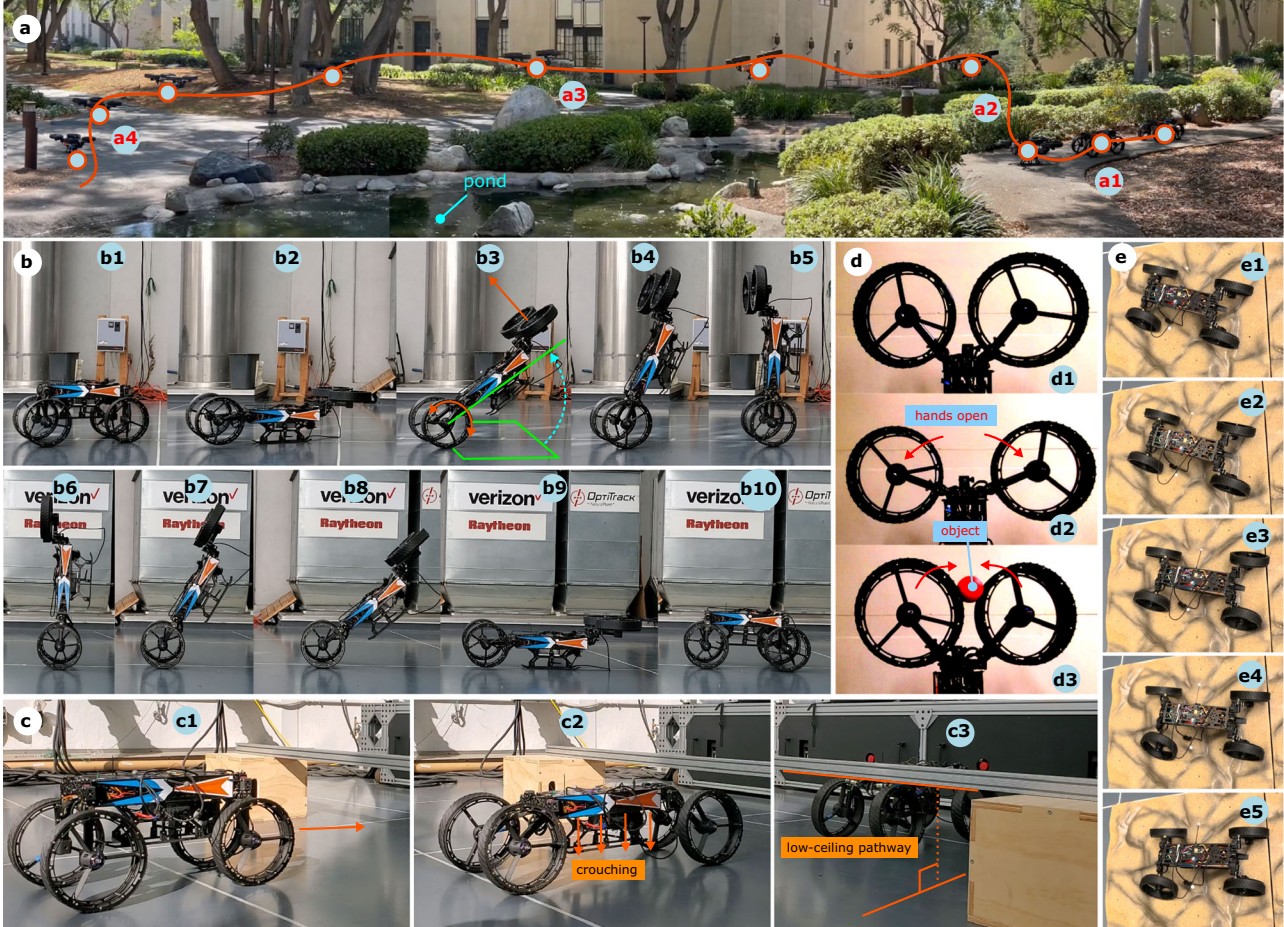

**Fig. 5 | M4's various locomotion modes. a** Ground-aerial locomotion near a pond. M4 rolls to the edge of the pond (a1), transforms into UAS mode and takes off (a2), flies over the pond to the opposite side (a3), and finally lands before transforming back into UGV mode (a4). **b** Illustrates the UGV to MIP and MIP to UGV maneuvers. M4 repurposes its front appendages to thruster mode (b1-b2), performs MIP maneuver to self-upright (b2-b4), dynamically balances in MIP mode (b5-b6),

descends in MIP maneuver (b7-b9), and finally transforms back into the UGV mode (b9-b10). **c** M4's crouching maneuver to pass under a low-clearance opening. **d** Shows M4's manipulation ability in MIP mode based on repurposing its free appendages. **e** M4 performs quadrupedal-legged locomotion on rough terrain by locking the wheels and translating the legs.

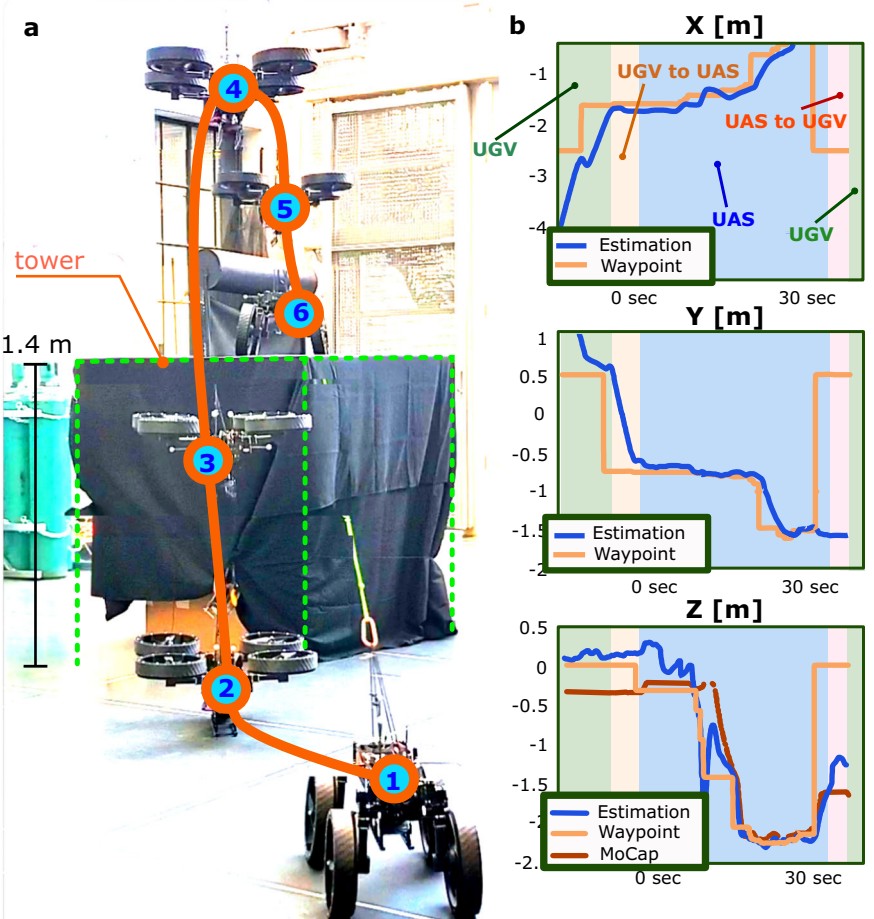

**Fig. 6 | Multi-modal probabilistic road map (MM-PRM) path planning with motion capture system. a** Shows the multi-modal path planning experiment performed inside a flight arena with motion capture cameras. In this experiment, the robot followed the path planning trajectory generated by the MM-PRM and A* algorithm to land on a 1.4 m tall platform. The robot started on the ground, drove to the UAS morphing location (1), transformed into the UAS mode (2), flew on top of the box (3 to 5), and transformed back into the UGV mode (6). **b** Illustrates the desired and actual trajectories.

desired angular rates of 10°/s and 5°/s were considered at the beginning (sit-down configuration) and near the end (stand-up configuration) during the uprighting phase (UGV to MIP transition), respectively. Similarly, the descending maneuver (MIP to UGV transition) uses the same controller.

M4 can perform crouching, object manipulation, and quadrupedal-legged locomotion as well (see Supplementary Video 1). As shown in Fig. 5c, the sagittal joints are employed to lower M4's main body to pass through low-ceiling pathways. Figure 5d shows that the free appendages (upper wheels) in MIP mode can be utilized for object manipulation purposes; however, the dexterity of object manipulation remains limited to grasping. Figure 5-e shows the M4's quadrupedal walking capability using quadrupedal-legged locomotion on rocky terrain. To perform quadrupedal locomotion, the wheels are locked. Then, the opposing legs are assigned swing and stance legs interchangeably. However, the absence of the knee joints in M4 prevents more natural gaits seen in other quadrupedal robots with more degrees of freedom in the legs.

We took two steps to show that M4 can operate in a fully self-sustained fashion. First, we designed a multi-modal path-planning algorithm and tested it using off-board sensing and computing (Fig. 6 based on Supplementary Video 6). Second, we translated this multi-modal path planner to onboard computers and sensors in M4 (see Fig. 7 based on Supplementary Video 7). Unlike the teleoperated pond tests, our experiments in the lab environment entailed autonomous multi-modal path planning and execution. The lab environment has an

OptiTrack motion capture (mocap) system. Several reflective markers were attached to the robot and environment. The mocap system's rigid body position and orientation measurements were transmitted to M4's computer through wireless communication. Then, a path-planning algorithm based on MM-PRM and A* algorithm steered the system towards the goal. The details and derivation of these algorithms can be seen in the Methods Section. Figure 6 shows one of the tests where M4 follows the calculated trajectory to land on top of a 1.4-m tall platform and transform back into UGV configuration. Then, we implemented this MM-PRM algorithm on the Jetson Nano computer on M4 to achieve fully autonomous and self-contained operation of M4 as shown in Fig. 7.

In the MIP maneuver (Fig. 5b), we demonstrated that M4 could repurpose its front and rear appendages to generate the external forces required to stand up and sit down entirely independently without external support. The maneuver provides two immediate mobility advantages: increased reach (or higher vantage point) and enhanced traction forces. The first advantage can be leveraged to tumble over large obstacles that cannot be handled with legged and wheeled mobilities. The second advantage can be employed to travel on steep slopes, similar to how birds use their wings and legs collaboratively to travel over inclined surfaces (i.e., WAIR maneuver). On these steep slopes, large traction forces are required. These forces cannot be substantiated by wheeled mobility.

The cartoon depictions of these maneuvers are shown in Fig. 8a and b. To perform the maneuver shown in Fig. 8a, the robot transforms

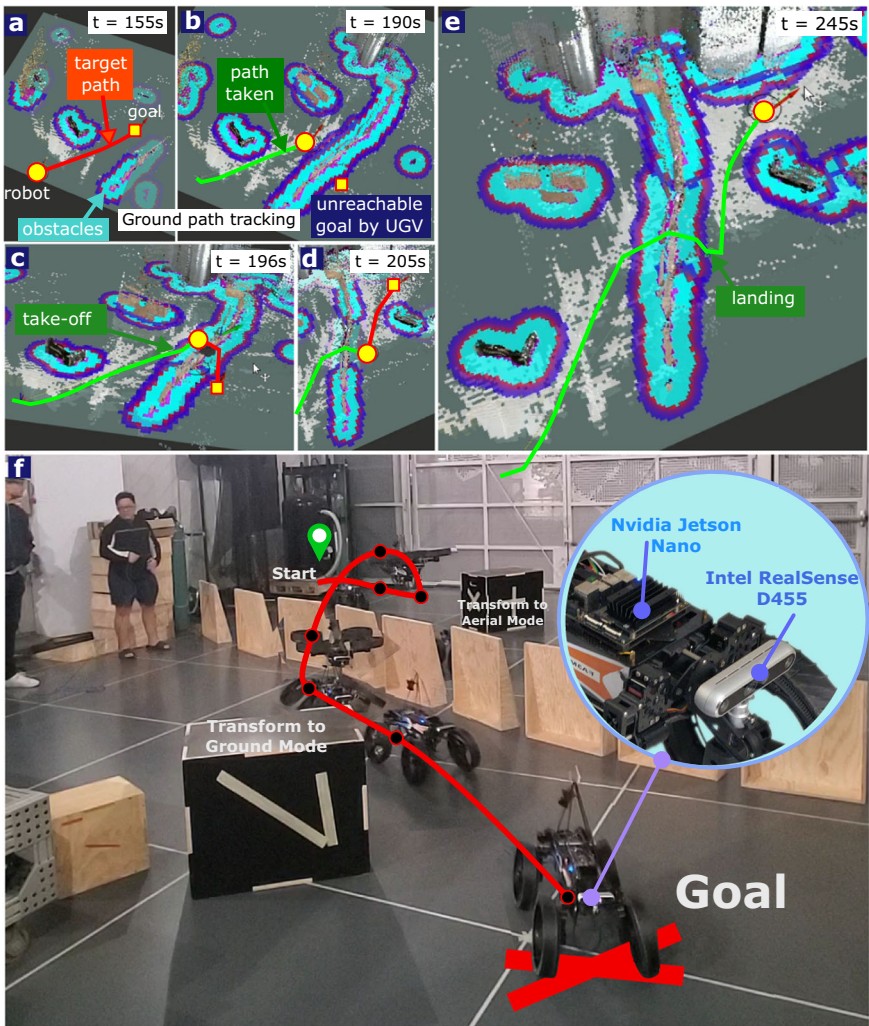

**Fig. 7 | Autonomous and self-contained MM-PRM path planning. a** Shows the online ground waypoint generation using MM-PRM and A* algorithm and the robot's ground trajectory tracking. The point cloud data captured by the Intel RealSense camera fixated in the front of M4 is processed in real-time by the Jetson Nano computer for autonomous path planning and navigation. **b** Illustrates a waypoint that is unreachable by the UGV mode. **c** Shows M4's autonomous transformation into UAS mode and flight over the obstacle to reach a desired waypoint. **d** Once the robot lands on the other side of the obstacles, it transforms into UGV mode and navigates towards the final waypoint. **e** Depicts the whole multimodal path taken by M4 from the start to goal point. **f** A composite image showing the path taken by the robot as it autonomously navigates the cluttered environment and switches from UGV to UAS modes.

into the MIP configuration. The upper thruster pushes the robot up the inclination while maintaining a certain pitch angle for stability. Then the robot changes back into UGV mode once the inclination has been cleared. Figure 8b illustrates the tumbling maneuver, where the robot uses its front or rear thrusters to lift one side of its body upwards and gain height advantage to clear a tall obstacle or vault over a large gap. First, M4 positions the front thrusters pointing upwards to lift the front side. Then, the rear wheels drive forward so the front side vault over the obstacle. M4 performs the same sequence with the rear thrusters and front wheels to fully clear the large obstacle.

As shown in Fig. 8c and d based on Supplementary Videos 3 and 4, we performed the WAIR and tumbling maneuvers in experiments. The WAIR, shown in Fig. 8c, was performed on a 45° upwards slope, which the robot cannot climb with the UGV or legged modes. The robot was initialized in the MIP configuration, then the wheel motors and thrusters worked together to propel the robot up the incline. The upper thrusters stabilized the robot's upper body tilt angle, and the wheel motors set the robot's forward speed on the slope. The tumbling maneuver, shown in Fig. 8d, utilized the same MIP uprighting maneuver shown in Fig. 5b to lift the robot's front side, drive forward, and vault over a large obstacle that the robot is unable to roll or walk over it

(Fig. 8d1 to Fig. 8d4). Then, the same maneuver was performed to lift the robot's backside, then, finally, the robot transformed back into UGV mode (Fig. 8d5 to Fig. 8d8).

## Discussions
We have presented M4 and showcased the advantages of considering morpho-functional appendages that can be repurposed to manipulate redundancy to enhance locomotion plasticity and achieve payload scalability. A few works that previously applied appendage repurposing in their designs achieved limited locomotion plasticity. Instead, in this paper, we demonstrated that our robot can (1) fly, (2) roll, (3) walk, (4) crouch, (5) balance, (6) tumble, (7) scout, and (8) loco-manipulate objects by switching the functionality of appendages between wheels, legs, hands, or thrusters. In addition, we demonstrated M4 can drive on steep slopes and vault over large obstacles if other modes were not applicable. We showed M4's design is scalable and can substantiate fully autonomous, self-contained, multi-modal operations. This modal diversity and level of autonomy have not been reported in multi-modal locomotion before and differentiates our robot from existing platforms.

The access to our wide array of actuators and locomotion modes allows the robot to choose the most efficient mode of locomotion

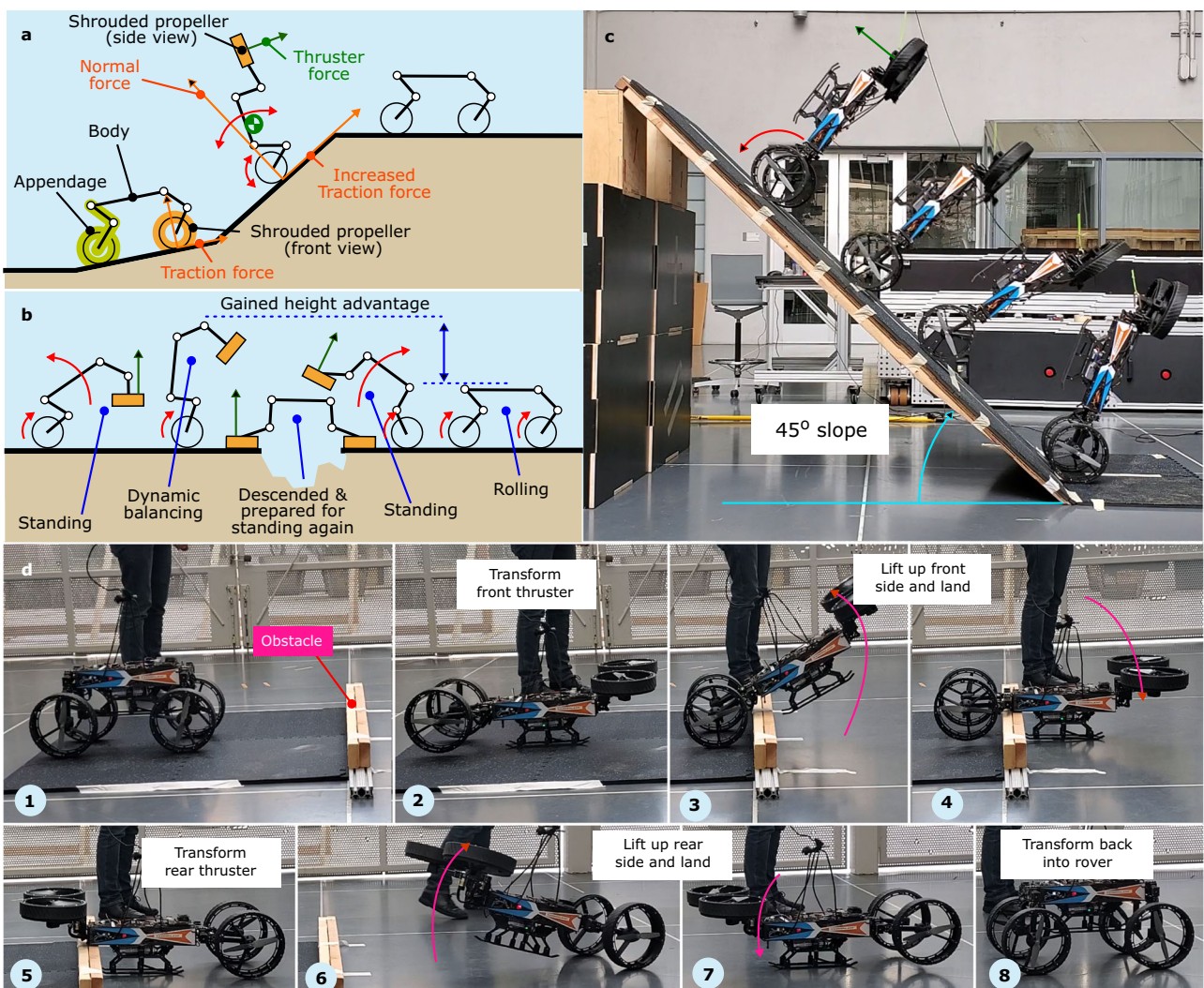

**Fig. 8 | Traversing over steep inclines and large obstacles. a** Illustrates M4's cartoon where the front appendages are repurposed to generate the external thrust forces to increase the ground reaction forces on a steep slope, a maneuver inspired by bird wing-assisted incline running (WAIR). **b** Demonstrates the various steps that are involved in M4's tumbling maneuver by repurposing its appendages. The achieved height advantage in the MIP mode is employed to overcome a large obstacle. **c** A composite image showing M4 uses MIP mode with thrusters to perform closed-loop WAIR on a 45° inclined surface. **d** Time-lapsed images of M4 performing a tumbling maneuver to vault over a large obstacle that cannot be handled by UGV or legged modes.

given the environment and obstacles. The energy cost of performing locomotion can be seen in Fig. 9, which lists the estimated electrical power consumed from the current load and reference voltages of all motors at a given environment and locomotion mode. Each wheel motor operates at 12V and draws 1–3A while each propeller motor operates at 24V and draws 20–40A, depending on the load. Each joint servo operates at 7.4V and draws 0.1–0.3A to transform or hold position, which consumes a relatively small power compared to the other motors. Therefore, it is significantly more energy efficient to use UGV locomotion and avoid using thrusters to save energy. Several modes that partially use thrusters, such as tumbling and thruster-assisted MIP, can consume less energy than UAS mode.

Our results suggest that redundancy manipulation using morphing appendages can present a powerful design view that not only can yield impressive locomotion plasticity within a single substrate but also can support crossing the boundaries of multi-substrate locomotion that involve conflicting requirements such as ground and air. We found that appendage repurposing is an effective tool for creating scalable designs when conflicting requirements exist. For instance, the increased thrust-to-weigh ratio achieved by repurposing all appendages to the thrusters in M4 can quadruple when all appendages are

repurposed to the thruster since the payload remains fixed. Remarkably, biologists reported these observations before; however, the robotic demonstrations remained unexplored or were not explored to the level showcased in this paper.

Future work will involve expanding M4's modes even further. For instance, dynamic legged locomotion gaits are a potential addition to existing capabilities. This goal can be achieved by increasing the number of degrees of freedom in the legs to support natural gaits. Currently, object manipulations during the MIP mode are limited to grasping. An exciting research path constitutes extending current manipulation capabilities to more complex scenarios such as holding tools. Also, from an autonomy standpoint, a decision-making algorithm to autonomously switch between all modes currently needs to be included. Currently, we can autonomously switch between UAS and UGV modes. We have augmented the M4 platform with a multi-modal MM-PRM path planning algorithm, Jetson Nano, and stereo depth camera from Intel RealSense, which is very light, efficient, and inexpensive, to create a point-cloud representation of the world in real-time; however, the decision-making algorithm needs further developments to be applicable in more complex scenarios. The addition of this level of autonomy allows M4 to create an occupancy map to evaluate the traversability of

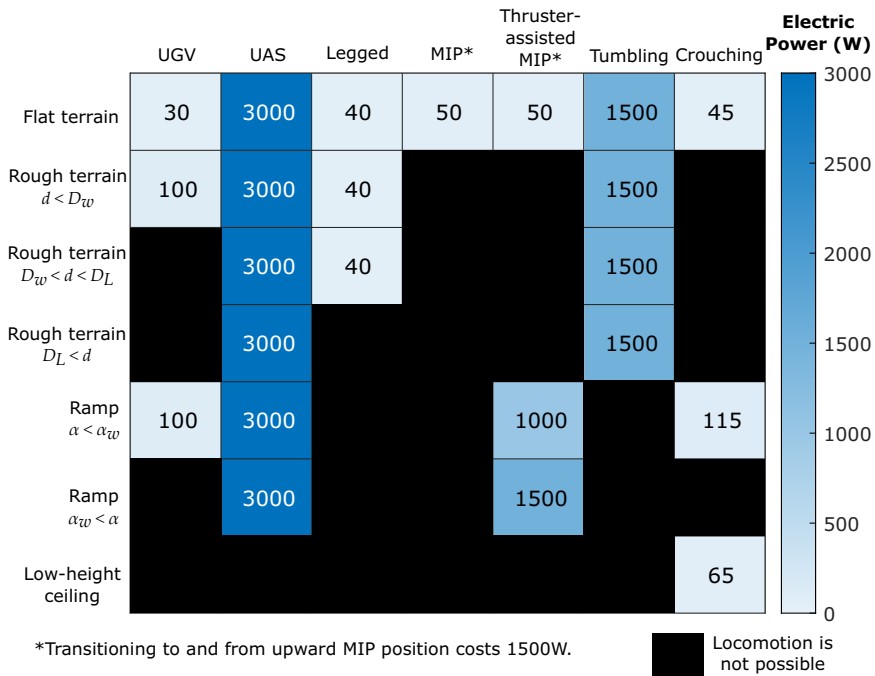

**Fig. 9 | Locomotion modes versus electrical power consumption.** The power consumption is estimated based on the average current and reference voltages during the locomotion. UGV is generally the most efficient mode of locomotion but is unable to traverse rough terrains, steep inclinations, or very large ground elevations. UAS mode offers the widest range of operation; however, it is the most energy-inefficient mode. The symbol $d$ denotes the obstacle size. $D_w$, and $D_L$ are the maximum obstacle sizes that can be handled with wheeled and legged locomotion, respectively. $\alpha_w$ denotes the maximum slope the UGV mode can handle.

the world for autonomous switching between different modes of locomotion. Notably, in this regard, the MIP configuration reported in this paper can be leveraged for scouting and enhancing the quality of created occupancy map. Finally, with the capabilities showcased in this paper and securing the missing capabilities, M4 can be employed in various applications, including search and rescue, space explorations, and package delivery to customers' doorsteps.

## Methods
### Brief overview of M4's dynamical modeling
In this section, we derive the dynamic model of the robot for control, analysis, and simulation. While there are many 3D simulation tools available (e.g., Gazebo, Simscape, MuJoCo), deriving the model symbolically to extract the inertia, Coriolis, and gravitational terms can be useful because it offers great flexibility for model-based nonlinear control design by presenting model components that can be programmed in M4 computer. Our modeling approach follows the Euler-Lagrangian equation of motion, where we first derive the conservative energy in the system of all massed components. Then, Lagrange formalism is applied. Figure 10 illustrates the free body diagram of the robot that shows the DOF and the robot's kinematics used throughout the modeling. M4 is composed of 13 rigid bodies (one main body, three linkages per leg) in our derivations.

Consider a set of massed components. For each $j$-th massed body, let $m_j$ denote the mass, $\mathbf{I_j} \in \mathbb{R}^{3 \times 3}$ be the principal inertia matrix, $\mathbf{p_j} \in \mathbb{R}^3$ and $\mathbf{v_j} \in \mathbb{R}^3$ be the inertial position and velocity vectors, respectively, and $\boldsymbol{\omega_j} \in \mathbb{R}^3$ be the angular velocity vector defined in the body frame of the $j$-th massed component. Furthermore, let $\mathbf{g} \in \mathbb{R}^3$ be the gravitational acceleration vector defined in the inertial frame. Then, the Lagrangian $\mathcal{L}$ is derived as the sum of the total kinetic and potential energy in the system and is given by:

$$\mathcal{L} = \sum_j \frac{1}{2}\left(m_j \mathbf{v_j}^\top \mathbf{v_j} + \widehat{\boldsymbol{\omega}}_\mathbf{j}^\top \mathbf{I_j} \widehat{\boldsymbol{\omega}}_\mathbf{j}\right) - m_j \mathbf{p_j}^\top \mathbf{g}, \quad (1)$$

where the first two terms are linear and angular kinetic energy, while the last is potential energy. In Eq. (1), the symbol $\widehat{\phantom{x}}$ denotes the skew symmetric operator.

Let $\mathbf{q} \in \mathcal{Q}$ ($\mathcal{Q}$ denotes the configuration variable space) be the generalized coordinates of the system, which consists of the body's 6 DOF (position and orientation), and the 2 DOF on each leg. Note that the shroud and propellers' angles are not part of the configuration variable vector. The equation of motion can then be derived using the Euler-Lagrangian formulation as follows:

$$\frac{d}{dt}\left(\frac{\partial \mathcal{L}}{\partial \dot{\mathbf{q}}}\right) - \frac{\partial \mathcal{L}}{\partial \mathbf{q}} = \mathbf{u_g} + \sum_k \left(\mathbf{u_{j,k}} + \mathbf{u_{w,k}} + \mathbf{u_{t,k}}\right), \quad (2)$$

where $\mathbf{u_{j,k}}$ and $\mathbf{u_{w,k}}$ are the generalized joint torques and wheel traction forces from $k$-th leg, respectively. And, $\mathbf{u_{t,k}}$ and $\mathbf{u_g}$ denote the generalized thruster and ground contact forces. The model given by Eq. (2) is highly generic. It can be considered for UGV, MIP, UAS, thruster-assisted MIP or WAIR, legged locomotion, and loco-manipulation in MIP mode. However, since MIP and WAIR involve active stabilization of M4 through collaborative thrust and traction wheel force regulations subject to contact force constraints, we decided to focus on these maneuvers only as they pose more technical control challenges compared to other maneuvers.

The ground forces are applied to each leg and the landing gear. The ground reaction forces are modeled using the Stribeck friction and compliant ground models from ref. 45. The compliant ground model uses springs and dampers with large stiffness and damping coefficients to calculate the normal forces. Then, (2) is written in the following state-space form:

$$\dot{\mathbf{x}} = \mathbf{f}(\mathbf{x}) + \mathbf{g}(\mathbf{x})\mathbf{u}, \quad (3)$$

where $\mathbf{x} \in \mathbb{R}^n$ and $\mathbf{u} \in \mathbb{R}^m$ denote the state and input vectors. The nonlinear terms $\mathbf{f}(\mathbf{x}), \mathbf{g}(\mathbf{x})$ embody all model terms, including gravity, inertial, and Coriolis matrices, shown in (2). The input vector $\mathbf{u}$

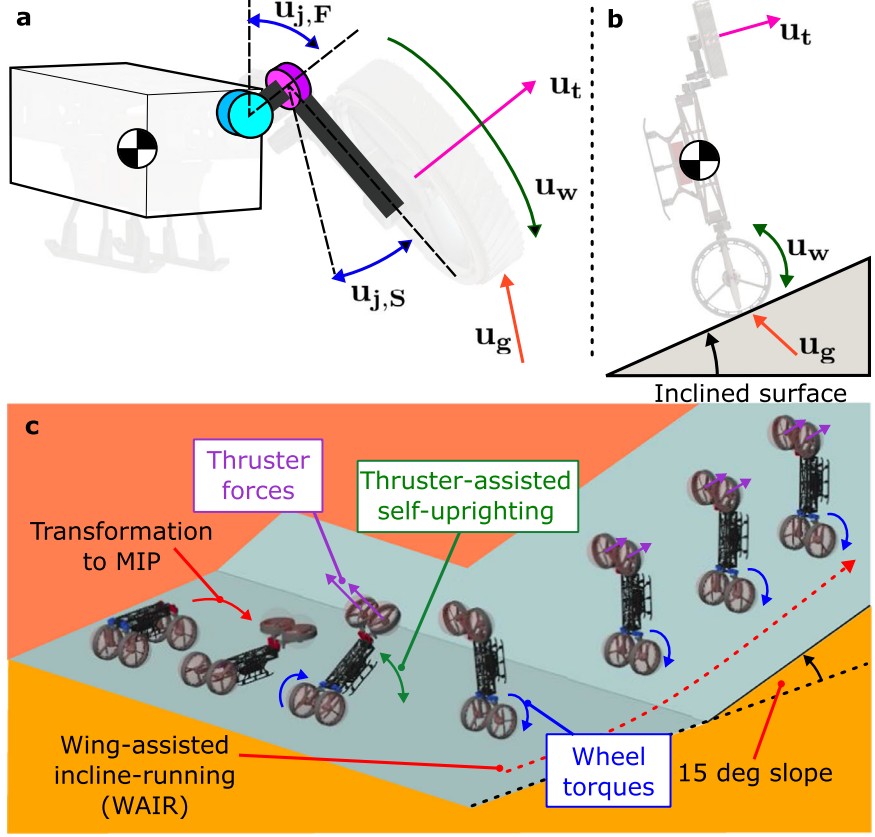

**Fig. 10 | Incline locomotion modeling and controls. a** The free-body diagram (FBD) of the robot in either UGV or UAS modes. Only one leg is shown for clarity. $\mathbf{u_{j,F}}$ and $\mathbf{u_{j,S}}$ are the frontal and sagittal joint torques, respectively. **b** FBD of the robot in MIP configuration as it climbs on a sloped surface and performs WAIR.

**c** Composite time-lapse illustration of the simulation, showing the transformation from UGV to MIP configuration, thruster-assisted self-uprighting, and WAIR on an inclined surface.

embodies thruster and joint actions as given in (2). Now, we apply the direct collocation method to resolve the MIP/WAIR problem.

### Direct collocation nonlinear dynamic programming

The approach we consider in this paper is based on the collocation technique. The main reason we consider the collocation technique for MIP control is the nature of the MIP problem. In MIP maneuver, M4 interacts with its environment through the traction forces generated at the contact points on the wheels and the thrust forces generated in the thrusters. The thrust forces are employed to manipulate the traction forces. If a closed-loop regulation of the thrust forces is not involved, then the required traction forces at the contact points cannot be achieved and slippage occurs. This problem can be formulated as an optimization-based control problem. Among available tools, collocation methods allow efficient dynamics approximation and constraint inclusion through interpolation functions. This property enables fast computation of control actions in real-time, which is important for highly dynamic maneuvers such as MIP/WAIR.

Consider $N$ time intervals during a gait cycle of the dynamic morphing systems given by:

$$0 = t_0 < t_1 < \ldots < t_N = t_f, \tag{4}$$

where $t_i$ denotes discrete times, $i = 0, \ldots, N$, and $t_f$ denotes the final discrete time. We consider the following cost function given by:

$$J = \sum_{i=0}^{N} (\mathbf{x_i} - \mathbf{x_{ref,i}})^\top \mathbf{Q} (\mathbf{x_i} - \mathbf{x_{ref,i}}) + \mathbf{u_i}^\top \mathbf{R} \mathbf{u_i}, \tag{5}$$

where $\mathbf{x_i} \in \mathbb{R}^n$ and $\mathbf{u_i} \in \mathbb{R}^m$ are the discrete-time states and inputs at time $t_i$, respectively, $\mathbf{x_{ref,i}}$ is the states reference, $t_N$ is the final time step, $\mathbf{Q}$ and $\mathbf{R}$ are weighting matrices. The cost $J$ is subject to a system of nonlinear equations given by (3) which is discretized with respect to time to obtain the following discrete-time system of equations:

$$\mathbf{x_{i+1}} = \mathbf{f_d}(\mathbf{x_i}) + \mathbf{g_d}(\mathbf{x_i})\mathbf{u_i}, \tag{6}$$

where $\mathbf{f_d}$ and $\mathbf{g_d}$ are the discretized system model from (3). We consider $2N$ boundary conditions given by:

$$\mathbf{r}(\mathbf{x_0}, \mathbf{x_N}, t_N) = \mathbf{0}, \quad \mathbf{r} \in \mathbb{R}^{2N}, \quad 0 \le t \le t_f, \tag{7}$$

which is derived from (6), in addition to $n_f$ inequality constraints describing the maximum input values and feasibility of contact forces based on friction-cone condition given by:

$$\mathbf{C}(\mathbf{x_0}, \ldots, \mathbf{x_N}, \mathbf{u_0}, \ldots, \mathbf{u_N}, t) \le \mathbf{0}, \quad \mathbf{C} \in \mathbb{R}^{n_f}, \quad 0 \le t \le t_f. \tag{8}$$

We stack all of the states and inputs from our system in the vectors $\mathbf{X} = [\mathbf{x_0}^\top, \ldots, \mathbf{x_N}^\top]^\top$ and $\mathbf{U} = [\mathbf{u_0}^\top, \ldots, \mathbf{u_N}^\top]^\top$. To approximate the discretized nonlinear dynamics from (3), we employ a method based on polynomial interpolation of states to simplify the computation of the cost function $J$.

Then, we stack $\mathbf{X}$ and $\mathbf{U}$ as the decision parameter vector $\mathbf{Y}$ for the nonlinear programming problem. Also, we add $t_f$ as the last entry of $\mathbf{Y}$,

resulting in the following decision parameter vector:

$$\mathbf{Y} = \left[ \mathbf{u_0}^\top, \ldots, \mathbf{u_N}^\top, \mathbf{x_0}^\top, \ldots, \mathbf{x_N}^\top, t_f \right]^\top. \tag{9}$$

At every sample time, we take the input to be as the linear interpolation function between $\mathbf{u_i}$ and $\mathbf{u_{i+1}}$ for $t_i \le t < t_{i+1}$:

$$\mathbf{u_{int}}(t) = \mathbf{u_i} + \frac{(t - t_i)}{(t_{i+1} - t_i)} \left( \mathbf{u_{i+1}} - \mathbf{u_i} \right). \tag{10}$$

We interpolate the states $\mathbf{x(t_i)}$ and $\mathbf{x(t_{i+1})}$ as well. However, we take a nonlinear cubic interpolation which is continuously differentiable and satisfies:

$$\dot{\mathbf{x}}_{\mathbf{int}}(s) := \mathbf{f}(\mathbf{x}(s)) + \mathbf{g}(\mathbf{x}(s))\mathbf{u}(s) = \mathbf{F}(\mathbf{x}(s), \mathbf{u}(s), s), \tag{11}$$

at $s = t_i$ and $s = t_{i+1}$. To do this, we write the following system of equations:

$$\begin{aligned}
\mathbf{x_{int}}(t) &= \sum_{k=0}^{3} \mathbf{h_k^i} \left( \frac{t - t_i}{\Delta t_i} \right)^k, \quad t_i \le t < t_{i+1}, \\
\mathbf{h_0^i} &= \mathbf{x_i}, \\
\mathbf{h_1^i} &= \Delta t_i \mathbf{F_i}, \\
\mathbf{h_2^i} &= -3\mathbf{x_i} - 2\Delta t_i \mathbf{F_i} + 3\mathbf{x_{i+1}} - \Delta t_i \mathbf{F_{i+1}}, \\
\mathbf{h_3^i} &= 2\mathbf{x_i} + \Delta t_i \mathbf{F_i} - 2\mathbf{x_{i+1}} + \Delta t_i \mathbf{F_{i+1}}, \\
\text{where } \mathbf{F_i} &:= \mathbf{F}(\mathbf{x_i}, \mathbf{u_i}, t_i), \quad \Delta t_i = t_{i+1} - t_i.
\end{aligned} \tag{12}$$

The interpolation function used for $\mathbf{x_{int}}$ must satisfy the derivatives at the discrete points $t_i$ and at the middle of sample times, that is, $t_{i+1}$. By inspecting Eq. (12), it can be seen that the derivative terms at the boundaries $t_i$ and $t_{i+1}$ are satisfied. Therefore, the only remaining constraints in the nonlinear programming constitute the collocation constraints at the middle $t_i \le t_{c,i} \le t_{i+1}$ time interval, the inequality constraints at $t_i$, and the constraints at $t_1$ and $t_f$. These constraints are given by:

$$\begin{aligned}
\mathbf{F}(\mathbf{x_{int}}(t_{c,i}), \mathbf{u_{int}}(t_{c,i}), t_{c,i}) - \dot{\mathbf{x}}_{\mathbf{int}}(t_{c,i}) &= \mathbf{0} \\
\mathbf{C}(\mathbf{x_{int}}(t_i), \mathbf{u_{int}}(t_i), t_i) &\le \mathbf{0} \\
\mathbf{r}(\mathbf{x_{int}}(t_0), \mathbf{x_{int}}(t_N), t_N) &= \mathbf{0}.
\end{aligned} \tag{13}$$

Simulations were performed in the Simscape environment where the robot transformed into the MIP configuration and performed WAIR on an inclined surface. Supplementary Fig. 3 shows the composite image and input-output plots of the simulation. Thrusters were utilized to stand up into the MIP configuration and assist in driving on an inclined surface with 15 degrees upwards slope. The simulation showed a stable transition to the MIP upright position and driving on the slope.

## Multi-Modal Probabilistic Road Map (MM-PRM) path planning

To take full advantage of the multi-modal capacities of our robot, it is necessary to develop path-planning optimization methods that work well with multiple modes of locomotion. Several works have already been done on multi-modal path planning for robots that can roll and fly such that the HyFDR[46,47] and the Drivocopter[35]. Most of the methods developed in these articles use a uniform discretization of the space, and then the optimal path is found with the Djikstra's algorithm[35], or with the A$^\star$[29,46]. Furthermore, in ref. 35, an optimization technique based on a reduced model of the system is used to calculate the costs of the edges and then to smooth the final trajectory. Araki et al.[29] have coupled their path planning method to a prioritization algorithm allowing swarm operation with 20 flying cars. While in ref. 47, Sharif

et al. have developed an algorithm to select the locomotion mode of the HyFDR robot allowing to optimize the cost of transport during outdoor navigation with only a 2D map of the environment.

The objective of path planning is to minimize the total energy consumed by the robot and optimize the choice of locomotion mode (ground or aerial). To achieve this goal, the environment is first discretized into a set of nodes where each node is associated with one of the locomotion modes. The nodes are then connected by edges and a cost for traveling between the nodes is computed. Finally, an A$^\star$ algorithm is used to determine the optimal path defined by a set of waypoints, each associated with a locomotion mode.

The 3D environment was discretized into more sparsely distributed points using the 3D MM-PRM algorithm. Like in ref. 48, this adapted version of the Probabilistic Road Map (PRM) algorithm takes into account the Multi-Modal nature of the robot's movements. The classical PRM algorithm builds a graph in the defined space by generating a certain number of nodes, where the nodes are created with random positions one by one. When a node is created, it will search for the nearest nodes already present in the graph and then connect to them to form edges while checking that it does not cross any obstacles. This method is adapted to generate a graph for unimodal robots by constraining the node generation to a single mode (i.e., create only ground nodes for the UGV mode or create aerial nodes for the UAV mode).

In this work, M4 can move both on the ground and in aerial space. Therefore, it is necessary to create 2 sets of constraints when generating the nodes. The main difference with the classical PRM algorithm is that a constraint is added on a certain number of nodes to ensure a sufficient number of nodes in each mode of locomotion. This extended version of the PRM algorithm requires the definition of 3 parameters: the number of ground surface nodes $N_w$, the number of nodes describing flyable space $N_f$, and the maximum distance between neighboring nodes $R$.

New ground nodes $\mathbf{p_{new}}$ are randomly assigned according to the following constraint:

$$\mathbf{p_{new}} \in \{(x, y, z) : z = z_g\}, \tag{14}$$

where $z_g$ is the ground elevation. Similarly, new nodes in the flyable task space are obtained as follows:

$$\mathbf{p_{new}} \in \{(x, y, z) : z > 0, z \ne z_g\}. \tag{15}$$

The search for neighboring nodes that will then be used to create the edges ($E$) is at the core of the PRM algorithm, and they are found using the following condition:

$$\mathbf{p_{nearest}} = \{\mathbf{p} \in \mathcal{N} : \| \mathbf{p_{new}} - \mathbf{p} \| \le R\}, \tag{16}$$

where $\mathcal{N}$ is the set of nodes already created, $R$ denotes the maximum radius distance, and $\|\cdot\|$ is the Euclidean norm.

The cost and time of calculation are very strongly linked to the choice of the values of the algorithm parameters ($R, N_w, N_f$). The greater the total number of nodes or the greater the radius of acceptance of the neighbors, the greater the computation time and cost will be. Therefore, it is necessary to study the convergence of the result in the function of the parameters to optimize to computation cost. We identified the parameters that led to the best results. The parameters are $R = 4$ meters, $N_w = 300$ and $N_f = 300$. An example of the graph built with the 3D MM-PRM algorithm is presented in Supplementary Fig. 2b.

To calculate the locomotion cost for the path planner, it is necessary to not only determine the costs associated with each mode but also the cost corresponding to the transition from one mode to another. As such, the cost of transport on a ground edge

denoted by $C_w$ is calculated using the power consumption by the wheel motor, $P_w$. Then, $P_w$ is integrated over the time of wheeled locomotion. The total power consumption is computed based on the torque and the angular velocity of each motor which is obtained from the current draw and encoder measurements. The time of ground locomotion is calculated based on the distance $d$ between the two nodes. As a result, $C_w$ is given by:

$$C_w = \int_0^{t_d} P_w(\tau)d\tau. \tag{17}$$

The energetic cost on a flying edge $C_f$ is computed using the power consumption $P_f$ in hovering, the robot forward velocity $v_f$ in flying mode, and the altitude $z$ of the two nodes. Hence, $C_f$ is given by:

$$C_f = P_f \frac{d}{v_f} + mg(z_2 - z_1), \tag{18}$$

where $z_1$ and $z_2$ are respectively the altitudes of nodes 1 and 2, $m$ is the mass of the robot, and $g$ is the gravitational acceleration constant. Last, the transition cost $C_t$ between the two modes is determined based on the power consumption of the joints during the morphing process $P_t$. Then, $P_t$ is integrated over the time of transition $t_t$ which yields:

$$C_t = \int_0^{t_t} P_t(\tau)d\tau. \tag{19}$$

These three energetic costs are employed to determine the optimal path in the edge space generated by the MM-PRM algorithm using the A* algorithm.

To find the optimal path in the graph, the A* path search algorithm[49] is used. The improved version of Dijkstra's algorithm[50] is employed to find the optimal path by using a heuristic function. The algorithm computes the best path to each node to only visit the most promising nodes. This avoids going through all possible paths and, therefore, finding the first-best optimal path with a low computational cost. Thus, each time the algorithm explores the $n$-th node, it calculates the minimum cost $f(n)$ necessary to reach the goal by passing through it using the following formula:

$$f(n) = g(n) + h(n), \tag{20}$$

where $g(n)$ is the real cost from the start to the n-th node, computed based on (21), and $h(n)$ denotes the heuristic cost to the goal. The heuristic cost $h(n)$ is calculated by summing two conservative costs. First, the cost of driving on flat ground to the goal in a straight line is calculated. Second, the cost of flying vertically along the $z$-axis to the goal is obtained. Since the cost of driving is much lower than flying, this is the most optimal way to move between two points if there are no obstacles or impassable terrains. The following cost for $g(n)$ is defined:

$$g(n) = \sum_{i=0}^{E_w} C_{w,i} + \sum_{j=0}^{E_f} C_{f,j} + N_t C_t, \tag{21}$$

where $E_w$ and $E_f$ are respectively the number of ground and aerial edges traveled by the robot, $C_{w,i}$ is the cost on the ground edge $i$, $C_{f,j}$ denotes the cost on the flying edge $j$, and $N_t$ represents the number of mode transition made by the robot.

Supplementary Fig. 2c shows the generated trajectories using the MM-PRM and A* algorithm to navigate three different environments.

Environments A and C require the robot to transition between ground and aerial modes to reach the target position, showing that the algorithm works for multi-modal applications.

## Data availability
Data will be provided upon request.

## Code availability
Simulation and path planning codes will be provided upon request.

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

## Acknowledgements

This project is funded by Caltech's Jet Propulsion Laboratory. A.R. efforts were partly funded by an NSF Foundational Research in Robotics (FRR), Award # 2142519, and a JPL Faculty Research Program (JFRP) fund. We acknowledge the work of graduate students and engineers at Caltech and Northeastern University for their help in prototyping and testing M4. Specifically, we are thankful to Dr. Milad Ramezani at Commonwealth Scientific and Industrial Research Organisation (CSIRO) and Filip Slezak from Swiss Federal Institute of Technology Lausanne (EPFL) for their help with autonomous multi-modal UAS-UGV operations. Noel Esparza-Duran supported the prototyping of M4. Benjamin Mottis from EPFL helped with preliminary path planning simulations and experiments.

## Author contributions

E.S. led the prototyping efforts, simulations, and experimentation. E.S. and A.R. collaboratively wrote the draft. A.K. evaluated the presented multi-modal models. R.N. supported prototyping efforts. A.K. and M.G. helped with draft editing. A.R. and M.G. conceived the M4 idea and are the principal investigators.

## Competing interests

The authors declare no competing interests.
