## [Peer Review File · Nature Communications]

REVIEWER COMMENTS

Reviewer #1 (Remarks to the Author):

This paper presents the design and control of a robot capable of performing many locomotion modes, such as flying, wheeling, walking, tumbling, and crouching. The design is mainly based on a mobile robot structure with wheels made from a propeller inside the wheels. Further, the wheels are connected to the body through a joint with two degree-of-freedom. Different locomotion modes are demonstrated, and algorithms to control the robot are also given.

The reviewer appreciates the efforts to design such a robot that probably has the greatest number of locomotion modes compared with existing robots. However, the reviewer does not find the proposed concept completely new, as quite a few recent works are using the same concept. Specifically, the paper lists three different views to achieve multimodal locomotion: 1) Morpho-Functionality; 2) Redundancy; 3) the combination of both. The paper then claims that the last view is unexplored. But there are quite a few works that belong to the third view, although they cannot achieve the many types of locomotion modes as demonstrated in this paper.

For the example shown in Fig. 2A, for which turtles and sea lions use their front flippers for swimming, and repurpose the same flippers to walk, see the recent cover article in Nature

Baines, Robert, Sree Kalyan Patiballa, Joran Booth, Luis Ramirez, Thomas Sipple, Andonny Garcia, Frank Fish, and Rebecca Kramer-Bottiglio. "Multi-environment robotic transitions through adaptive morphogenesis." *Nature* 610, no. 7931 (2022): 283-289.

For the example shown in Fig. 2C, see the following work (although this work didn't have quadrupedal locomotion, it did repurpose the wing to walk on the ground):

Daler, Ludovic, Stefano Mintchev, Cesare Stefanini, and Dario Floreano. "A bioinspired multi-modal flying and walking robot." *Bioinspiration & biomimetics* 10, no. 1 (2015): 016005.

From the reviewer's perspective, the presented work is a combination of a wheeled and legged robot (e.g., Mars Rover or the robot shown in the following paper) and a flying robot with the propellers attached to the wheel (Flying star as cited in the paper). The wheeled and legged robot can have the following modes: crawling, wheeling, tumbling, and crouching, while the flying star can have: flying, wheeling, and crouching.

Bjelonic, Marko, Prajish K. Sankar, C. Dario Bellicoso, Heike Vallery, and Marco Hutter. "Rolling in the deep—hybrid locomotion for wheeled-legged robots using online trajectory optimization." *IEEE Robotics and Automation Letters* 5, no. 2 (2020): 3626-3633.

Besides the major concerns, the following are several issues related to the organizations and presentations for the paper.

How are the three views related to the overarching question stated at the beginning: Which design views yield scalable robots with large locomotion plasticity? This question is proposed but never answered.

The tumbling locomotion mode is only demonstrated on flat ground. But the paper claims the advantage of tumbling is for overcoming slopes as shown in Fig. 7A, which is only an illustration. Simulation results are provided in Fig. 9 by theoretical modeling of the dynamics and a dynamic programming method, but why can experiments not be done? Note that standing up with two wheels can also be done for a wheeled-legged robot. Similarly, the tumbling maneuver is not clearly demonstrated with only an illustration in Fig. 7B.

The reviewer thinks the first section 1 is too long. It should be shortened to clearly demonstrate the contributions. Using biology as examples (section 1.2) should be ok, but some brief discussions are enough. Fig. 2E is only hypothetical, which was not realized. Maybe it can be included in the supplementary material. The state-of-art is also quite long, for which the pictures can be moved to the supplementary material.

The reviewer thinks section 2 should be expanded, especially an overview of the design of the robot. Basic degree-of-freedom should be discussed in the main texts even though it is straightforward in Fig. 4B (e.g., swing motion, sideways motion). Also, it might be better to briefly describe how to achieve each different locomotion modes by properly repurposing the legs and actuating the motors (For instance, some detailed descriptions can be provided for the repurposing in Fig. 4A).

The paper should briefly discuss why the specific direct collocation nonlinear dynamic programming is used for solving the WAIR problem. Also, it seems the whole modeling is only used for this WAIR. Can the model be used for any other locomotion modes?

The reviewer thinks Fig. 11 does not add much information to the paper.

Reviewer #2 (Remarks to the Author):

This is an interesting work proposed by Sihite et al. about the implementation of a multi-modal mobility robot featuring non-less than seven mobility modes. However, I have the following concerns that prevent the paper to be accepted for publication: the main part of the work has been put on the design and implementation of the mechatronics as well as the control of the robot. This work is really impressive. However, the results are not really at the level of a journal like Nature Communications for the following reasons :

- the relationship to biology and the biological plausibility of the robotic implementation is not very convincing as the final solution (a quadrotor with tilted propellers) is very far from solutions used for example by birds for implementing multi-modal locomotion.
- the state-of-the-art is very exhaustive but it is the kind we find more in review paper rather than in research article.
- one of the most interesting scenario that could clearly show the great interest in developing such multi-modal mobility was only simulated and it is poorly depicted in figure 7. I strongly suggest authors to implement this scenario physically.
- multi-modal mobility can also find a great interest in energy saving but even this point is poorly illustrated.

According to the work and the results obtained, I can not recommend this work for a publication in Nature Communications.

Reviewer 1 Comments

- The reviewer appreciates the efforts to design such a robot that probably has the greatest number of locomotion modes compared with existing robots. However, the reviewer does not find the proposed multi-modal design completely new, as quite a few recent works are using the same concept.

Authors' response: You make a fair assessment. We extended our experimental results based on your suggestions. We kindly invite you to assess our robot's new capabilities, including locomotion on steep slopes (45 deg.) and over large obstacles. In addition, now, M4 can realize fully autonomous multi-modal locomotion using its onboard computer and sensors in a self-contained fashion.

In this revised draft, the old Experimental Results Section (page 13, line 24) has been replaced by a new version in page 15, line 24 that covers our new tests.

- Specifically, the paper lists three different views to achieve multimodal locomotion: 1) Morpho-Functionality; 2) Redundancy; 3) the combination of both. The paper then claims that the last view is unexplored. But there are quite a few works that belong to the third view, although they cannot achieve the many types of locomotion modes as demonstrated in this paper. For the example shown in Fig. 2A, for which turtles and sea lions use their front flippers for swimming, and repurpose the same flippers to walk, see the recent cover article in Nature Baines, Robert, Sree Kalyan Patiballa, Joran Booth, Luis Ramirez, Thomas Sipple, Andonny Garcia, Frank Fish, and Rebecca Kramer-Bottiglio. "Multi-environment robotic transitions through adaptive morphogenesis." Nature 610, no. 7931 (2022): 283-289.

Authors' response: Thanks for your comment. We admit Baines et al. work follows the principles of View 3. We cited this work in page 7, line 37, then we explained how M4 differs from this amphibious system.

M4 differs from Baines' work because Baines' robot does not manipulate appendage redundancy through morphing to maximize locomotion plasticity.

Specifically speaking, Baines' robot repurposes:

- four flippers (swimming)
- four flippers into four legs (walking)
- other possible repurposing but NOT pursued in Baines' work
 - * four flippers into two legs (scouting)
 - * four flippers into two legs (walking)
 - * four flippers into two legs + two hands (loco-manipulation)
 - * four flippers into four hands (dexterous manipulation)
 - * four flippers into two legs + two wings (WAIR/MIP)
 - * four flippers into two wings (flight)

* four flippers into four wings (high thrust-to-weight ratio flight)

(Note that not all of the items listed above are realizable by amphibious animals that inspire Baines work; however, they are listed to show all of the modes.) Instead, M4 repurposes:

- four legs (quadrupedal-legged locomotion)
- four legs to four thrusters (flight)
- four legs to two thrusters + two wheels (WAIR over 45-deg slopes)
- four legs to two thrusters + two wheels (tumble over large obstacles)
- four legs to two wheels + two hands (loco-manipulation in MIP)
- four legs to two wheels (MIP)
- four legs to four wheels (UGV)

M4's redundancy manipulation is not reported in Baines' paper.

- For the example shown in Fig. 2C, see the following work (although this work didn't have quadrupedal locomotion, it did repurpose the wing to walk on the ground): Daler, Ludovic, Stefano Mintchev, Cesare Stefanini, and Dario Floreano. "A bioinspired multi-modal flying and walking robot." *Bioinspiration & biomimetics* 10, no. 1 (2015): 016005.

Authors' response: M4 and Daler both can fly and walk. Daler repurposes:

- two wings into two legs (walking)

Daler does not possess the M4's redundancy manipulation abilities. Daler's ground mobility and payload capacity are minimal. Daler cannot host computers and sensors for autonomy.

- From the reviewer's perspective, the presented work is a combination of a wheeled and legged robot (e.g., Mars Rover or the robot shown in the following paper) and a flying robot with the propellers attached to the wheel (Flying star as cited in the paper). The wheeled and legged robot can have the following modes: crawling, wheeling, tumbling, and crouching, while the flying star can have: flying, wheeling, and crouching.

Authors' response: You make a fair assessment. M4 and Flying Star both can fly and roll (wheel mobility). However, Flying Star repurposes:

- four wheels into four thrusters

Other possible repurposings were not pursued because Flying Star possesses

- one actuator to transform its body shape whereas M4 possesses eight actuators
- two actuated wheels whereas M4 has four actuated wheels
- separated wheels and thrusters whereas M4 possesses integrated wheels and thrusters

Flying Star presents a design that does not match M4’s complexity in terms of the number of actuated joints, hardware design, low-level closed-loop control, and high-level autonomy.

- Bjelonic, Marko, Prajish K. Sankar, C. Dario Bellicoso, Heike Vallery, and Marco Hutter. ”Rolling in the deep–hybrid locomotion for wheeled-legged robots using online trajectory optimization.” *IEEE Robotics and Automation Letters* 5, no. 2 (2020): 3626-3633.

Authors’ response: This is an excellent comment. We admit this work belongs to View 3. We cited Bjelonic’s work in page 7, line 37. Bjelonic’s robot and M4 can roll, MIP, and walk (quadruped).

Bjelonic’s work repurposes

- four legs (quadrupedal locomotion)
- four legs into two legs + two wheels (MIP mode)

Bjelonic’s work does not possess the M4’s redundancy manipulation abilities. MIP ability showcased in Bjelonic’s work dose not match M4’s MIP ability. M4 climbs up 45-deg. steep slopes and tumbles over obstacles in MIP mode owing to its thrusters.

- How are the three views related to the overarching question stated at the beginning: Which design views yield scalable robots with large locomotion plasticity? This question is proposed but never answered.

Authors’ response: Thanks for raising this concern. We modified the text to address your concern as follows.

In multi-modal locomotion, the conflicting design requirements for each mode negatively affect the mobility in other modes, causing scalability issues. We explained the scalability issue at page 4, line 10. This scalability issue grows when high locomotion plasticity is pursued. We identified three mainstream views in the multi-modal robot literature. We explained these views at

- View 1: morpho-functional design (see page 5, line 3), e.g., two wings turned to two legs
- View 2: redundant design (see page 6, line 20), e.g., two wing + two legs
- View 3: design based on morpho-functionality utilized to manipulate redundancy (see page 6, line 43), e.g., four wings into
 - * four legs (large support contact force)
 - * two legs + two wings (large traction force)
 - * two legs + two hands (loco-manipulation)
 - * two legs (scouting)
 - * etc.

Then, we identified evidence from animal studies that View 3 is employed by animals as a mechanism to combat scalability issues [1]. Last, we answered

the question 'Which design views yield scalable robots with large locomotion plasticity?' in page 12, line 1.

- Comments related to tumbling and WAIR (Figure 7 and Figure 9). The tumbling locomotion mode is only demonstrated on flat ground. The paper claims the advantage of tumbling is for overcoming slopes as shown in Fig. 7A, which is only an illustration. Simulation results are provided in Fig. 9 by theoretical modeling of the dynamics and a dynamic programming method, but why can experiments not be done? The tumbling maneuver is not clearly demonstrated with only an illustration in Fig. 7B.

Authors' response: You are fair in your assessment. We addressed your concern by experimentally validating tumbling and incline climbing concepts (page 20, line 18).

- The reviewer thinks the first section 1 is too long. It should be shortened to clearly demonstrate the contributions.

Authors' response: We addressed your concern as follows.

- We completely removed Design Rationale (page 8, line 15) and State Of The Art (page 9, line 29) Subsections from the old draft's Introduction Section.
 - We incorporated a shortened version of the State Of The Art Subsection in a Subsection called Objectives and Design Views in page 5, line 3, page 6, line 20, and page 6, line 43.
 - To better convey the paper's contributions, we placed Paper Contributions in page 2, line 37 in Section 1.
- Using biology as examples (section 1.2) should be ok, but some brief discussions are enough.

Authors' response: To address your concern, we removed the Design Rationale (page 8, line 15). We moved Fig. 2-E to the supplementary materials (Fig. 11). We incorporated a shortened version of the biology examples to View 3 (page 6, line 43) in the Objectives and Design Views in the revised draft.

- Fig. 2E is only hypothetical, which was not realized. Maybe it can be included in the supplementary material.

Authors' response: We moved Fig. 2-E to the supplementary materials (Fig. 11).

- The state-of-art is also quite long, for which the pictures can be moved to the supplementary material.

Authors' response: We addressed your concern as follows.

- We removed the State Of The Art Subsection (page 9, line 29).
 - Then, we incorporated shortened segments of the removed State Of The Art Subsection in a Subsection called Objectives and Design Views in page 5, line 3, page 6, line 20, and page 6, line 43.
 - We placed the literature review composite picture in the supplementary materials (Fig. 10).
- The reviewer thinks section 2 should be expanded, especially an overview of the design of the robot. Basic degree-of-freedom should be discussed in the main texts even though it is straightforward in Fig. 4B (e.g., swing motion, sideways motion).

Authors' response: We addressed this concern as follows.

- Based on your concern, we expanded the Design Overview Subsection (page 12, line 17) in the revised draft. In Design Overview Subsection, we discussed basic degrees of freedom in page 12, line 23.
 - We added a new subsection in Section 2 of the revised draft called Design Rationale page 11, line 32. Design Rationale Subsection in the revised draft covers an overview of M4's design philosophy and which view was adopted.
 - We expanded Results Subsection based on new tests.
- Also, it might be better to briefly describe how to achieve each different locomotion modes by properly repurposing the legs and actuating the motors (For instance, some detailed descriptions can be provided for the repurposing in Fig. 4A).

Authors' response: Thank you for this comment. In page 12, line 23, we addressed your request and added detailed descriptions on the repurposing shown in Fig. 4-A.

- The paper should briefly discuss why the specific direct collocation nonlinear dynamic programming is used for solving the WAIR problem. Also, it seems the whole modeling is only used for this WAIR. Can the model be used for any other locomotion modes?

Authors' response: Thank you for this suggestion. In page 25, line 35, we explained why the collocation method is used. In page 25, line 16, we explained if the model can be used for other modes and why we focused only on WAIR in the Methods Section.

- The reviewer thinks Fig. 11 does not add much information to the paper.

Authors' response: We removed Fig. 11.

Reviewer 2 Comments

This is an interesting work proposed by Sihite et al. about the implementation of a multi-modal mobility robot featuring non-less than seven mobility modes. However, I have the following concerns that prevent the paper to be accepted for publication: the main part of the work has been put on the design and implementation of the mechatronics as well as the control of the robot. This work is really impressive. However, the results are not really at the level of a journal like Nature Communications for the following reasons:

- the relationship to biology and the biological plausibility of the robotic implementation is not very convincing as the final solution (a quadrotor with tilted propellers) is very far from solutions used for example by birds for implementing multi-modal locomotion.

Authors' response: We appreciate your feedback. In the design of M4, we are focused on animals' strategies to enhance locomotion plasticity (manipulate appendage redundancy through morphing) rather than the shape of their appendages (flapping versus rotary wings).

You make a fair assessment. The connection between M4 and biology was not clear in the initial draft. We delineated this point in page 12, line 1.

- the state-of-the-art is very exhaustive but it is the kind we find more in review paper rather than in research article.

Authors' response: Thanks for your feedback. We removed the State Of The Art Subsection. We incorporated a shortened version of it in the Objectives and Design Views (page 4, line 8).

- one of the most interesting scenario that could clearly show the great interest in developing such multi-modal mobility was only simulated and it is poorly depicted in figure 7. I strongly suggest authors to implement this scenario physically.

Authors' response: You make a fair assessment. We extended our experimental results based on your suggestions. We kindly invite you to assess our robot's new capabilities, including locomotion on steep slopes (45 deg.) and over large obstacles. In addition, now, M4 can realize fully autonomous multi-modal locomotion using its onboard computer and sensors in a self-contained fashion.

In this revised draft, the old Experimental Results Section (page 13, line 24) has been replaced by a new version in page 15, line 24 that covers our new tests.

- multi-modal mobility can also find a great interest in energy saving but even this point is poorly illustrated.

Authors' response: We have added Fig. 9 which lists the estimated power consumption of the robot under various terrain and locomotion modes to clarify the difference in power consumption. A discussion on the estimated power consumption can be seen in page 22, line 1.

REVIEWERS' COMMENTS

Reviewer #1 (Remarks to the Author):

The reviewer appreciates the authors' efforts in revising the paper and acknowledges the improvements made in comparison to the previous version. However, the reviewer remains unconvinced about the novelty of the developed system, as similar concepts have been previously presented. While the reviewer recognizes the impressive number of locomotion modes that the developed robot possesses, the design primarily demonstrates remarkable engineering work without introducing groundbreaking ideas for future research.

Moreover, the paper's main question revolves around scalability, which is defined in the paper as "a mobile robot design is scalable if its payload capacity can be increased such that its mobility is not severely affected." This definition is difficult to quantify. The authors contend that other designs lack scalability, but others do not specifically test whether their robots' mobility is affected by how much when a payload is added. Furthermore, even for the design presented in this paper, the authors have not experimented with increasing the payload to determine its impact on mobility.

Reviewer #2 (Remarks to the Author):

Authors have addressed correctly all my concerns. The paper can now be accepted for publication.

Reviewer 1's Comments

The reviewer appreciates the authors' efforts in revising the paper and acknowledges the improvements made in comparison to the previous version. However, the reviewer remains unconvinced about the novelty of the developed system, as similar concepts have been previously presented.

- While the reviewer recognizes the impressive number of locomotion modes that the developed robot possesses, the design primarily demonstrates remarkable engineering work without introducing ground-breaking ideas for future research.

Authors' response: Thanks for your comment. In 'Paper Contributions' and 'Design Rationale' Sections, we introduced the paper's main idea based on appendage redundancy manipulation through structure repurposing to maximize locomotion plasticity and combat scalability issues in multi-modal robots.

We tested our claims by introducing M4's hardware, locomotion control design, and autonomy. In our culminating tests, we delivered extensive locomotion plasticity with a scalable design (approx. 3 kg payload capacity and 6 kg body weight). We invite you to review a comparison between M4 and the state-of-the-art multi-modal robots listed below in terms of scalability-plasticity relationship to evaluate the potentials of the main idea in transforming future mobile robot research:

- **Swarm robot** (Araki et al., 2017) with **two modes**, weighs 40 g and can generate approx. 70 g thrust force yielding 30-g payload capacity, **one-hundredth of M4's payload capacity**.
 - **Cobots** (Tagliabue et al., 2020) with **two modes**, weighs 800 g and can generate approx. 1000 g thrust force yielding a predicted 200-g maximum payload capacity, **one-fifteenth of M4's payload capacity**.
 - **Leonardo** (Kim et al., 2021) with **two modes**, weighs 2600 g and can generate approx. 2800 g thrust force yielding a predicted 200-g maximum payload capacity, **one-fifteenth of M4's payload capacity**.
 - **FlyingStar** (Meiri et al., 2019) with **two modes**, weighs 900 g and can generate approx. 1300 g thrust force yielding a predicted 400-g maximum payload capacity, approx. **one-seventh of M4's payload capacity**.
 - **M4 (this work)** with **eight modes**, weighs 6000 g and can generate approx. 9000 g thrust force yielding a **3000-g maximum payload capacity**.
- Moreover, the paper's main question revolves around scalability, which is defined in the paper as "a mobile robot design is scalable if its payload capacity can be increased such that its mobility is not severely affected."

- This definition is difficult to quantify.

Authors' response: While there are various ways to measure scalability, one fundamental approach is to evaluate it based on the maximum allowable payload that the system can carry before it becomes completely immobilized in any mode. Based on your comment, we added a definition of quantifying scalability in page 3, line 14.

- The authors contend that other designs lack scalability, but others do not specifically test whether their robots' mobility is affected by how much when a payload is added.

Authors' response: Your assessment is accurate. We assert other designs lack scalability because these examples with 2-3 modes possess limited payload capacity. And, if the number of locomotion modes was increased beyond 2-3 modes, these redundant designs would face even more significant scalability issues due to two main reasons:

Firstly, when the number of locomotion modes in these redundant examples increases, the total mass of the system will be the sum of the mass introduced by each mode. The added mass for each mode includes the component mass from actuators, sensors, power electronics, and other devices used in that particular mode.

Secondly, in addition to the added mass from each mode, there is another form of added mass that must be considered: the added mass required to avoid the risk of immobilization. As the mass from other modes accumulates, some modes (such as the UAS and Legged modes) require the addition of large actuators, power electronics, and batteries to prevent the risk of immobilization. In other words, in these modes, component size (and mass) rapidly increases as the total mass increases. Other modes may be less sensitive to mass increase. For example, the manipulation mode is not affected by an increase in the total mass since it depends solely on the object's mass, not the robot's mass. And, the wheeled mode is less sensitive to mass increase compared to the legged mode, as joint actuators in the legged mode bear weight, while wheels only produce traction forces.

By repurposing the appendages in M4 (paper's main idea), the mass from each mode can be shared with other modes, resulting in a reduced total mass and smaller components in sensitive modes.

We added a summary of this response to page 6, line 15.

- Furthermore, even for the design presented in this paper, the authors have not experimented with increasing the payload to determine its impact on mobility.

Authors' response: We tested the scalability of M4 by adding onboard computers and exteroceptive sensors and demonstrated that it is capable of performing eight different locomotion modes while supporting a total payload capacity of 3000 g. This is a noteworthy improvement over previous works that did not report such payload capacities and locomotion plasticity.

Reviewer 2's Comments

Authors have addressed correctly all my concerns. The paper can now be accepted for publication.